# High-Dimensional Bayesian Optimization via Nested Riemannian Manifolds

**Noémie Jaquier**[1,2]
[1]Idiap Research Institute
1920 Martigny, Switzerland
noemie.jaquier@kit.edu

**Leonel Rozo**[2]
[2]Bosch Center for Artificial Intelligence
71272 Renningen, Germany
leonel.rozo@de.bosch.com

## Abstract

Despite the recent success of Bayesian optimization (BO) in a variety of applications where sample efficiency is imperative, its performance may be seriously compromised in settings characterized by high-dimensional parameter spaces. A solution to preserve the sample efficiency of BO in such problems is to introduce domain knowledge into its formulation. In this paper, we propose to exploit the geometry of non-Euclidean search spaces, which often arise in a variety of domains, to learn structure-preserving mappings and optimize the acquisition function of BO in low-dimensional latent spaces. Our approach, built on Riemannian manifolds theory, features geometry-aware Gaussian processes that jointly learn a nested-manifold embedding and a representation of the objective function in the latent space. We test our approach in several benchmark artificial landscapes and report that it not only outperforms other high-dimensional BO approaches in several settings, but consistently optimizes the objective functions, as opposed to geometry-unaware BO methods.

## 1 Introduction

Bayesian optimization (BO) is considered as a powerful machine-learning based optimization method to globally maximize or minimize expensive black-box functions [50]. Thanks to its ability to model complex noisy cost functions in a data-efficient manner, BO has been successfully applied in a variety of applications ranging from hyperparameters tuning for machine learning algorithms [51] to the optimization of parametric policies in challenging robotic scenarios [11, 16, 41, 49]. However, BO performance degrades as the search space dimensionality increases, which recently opened the door to different approaches dealing with the curse of dimensionality.

A common assumption in high-dimensional BO approaches is that the objective function depends on a limited set of features, i.e. that it evolves along an underlying low-dimensional latent space. Following this hypothesis, various solutions based either on random embeddings [57, 43, 9] or on latent space learning [13, 23, 42, 59] have been proposed. Although these methods perform well on a variety of problems, they usually assume simple bound-constrained domains and may not be straightforwardly extended to complicatedly-constrained parameter spaces. Interestingly, several works proposed to further exploit the observed values of the objective function to determine or shape the latent space in a supervised manner [59, 42, 4]. However, the integration of *a priori* domain knowledge related to the parameter space is not considered in the learning process. Moreover, the aforementioned approaches may not comply easily to recover query points in a complex parameter space from those computed on the learned latent space.

Other relevant works in high-dimensional BO substitute or combine the low-dimensional assumption with an additive property, assuming that the objective function is decomposed as a sum of functions of low-dimensional sets of dimensions [33, 37, 21, 44, 24]. Therefore, each low-dimensional partition

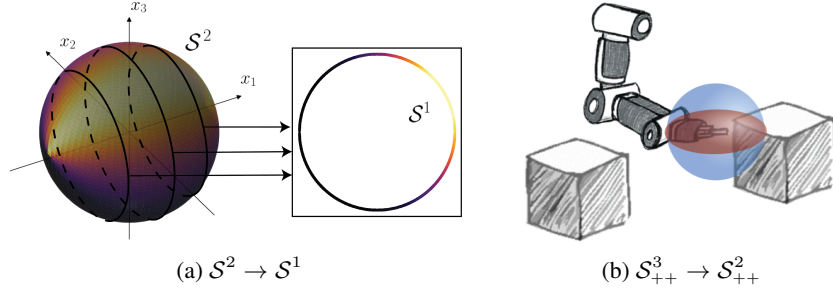

(a) $\mathcal{S}^2 \rightarrow \mathcal{S}^1$        (b) $\mathcal{S}^3_{++} \rightarrow \mathcal{S}^2_{++}$

Figure 1: Illustration of the low-dimensional assumption on Riemannian manifolds. (*a*) The function on $\mathcal{S}^2$ is not influenced by the value of $x_1$ and may be represented more efficiently on the manifold $\mathcal{S}^1$. (*b*) The stiffness matrix of a robot is optimized to push objects lying on a table. As the stiffness along the axis $x_3$ does not influence the pushing skill, the cost function may be better represented in a latent space $\mathcal{S}^2_{++}$. Note that the manifolds dimensionality is limited here due to the difficulty of visualizing high-dimensional parameter spaces. However, these examples are extensible to higher dimensions.

can be treated independently. In a similar line, inspired by the dropout algorithm in neural networks, other approaches proposed to deal with high-dimensional parameter spaces by optimizing only a random subset of the dimensions at each iteration [36]. Although the aforementioned strategies are well adapted for simple Euclidean parameter spaces, they may not generalize easily to complex domains. If the parameter space is not Euclidean or must satisfy complicated constraints, the problem of partitioning the space into subsets becomes difficult. Moreover, these subsets may not be easily and independently optimized as they must satisfy global constraints acting on the parameters domain.

Introducing domain knowledge into surrogate models and acquisition functions has recently shown to improve the performance and scalability of BO [11, 3, 45, 30, 14]. Following this research line, we hypothesize that building and exploiting geometry-aware latent spaces may improve the performance of BO in high dimensions by considering the intrinsic geometry of the parameter space. Fig. 1 illustrates this idea for two Riemannian manifolds widely used (see § 2 for a short background). The objective function on the sphere $\mathcal{S}^2$ (Fig. 1a) does not depend on the value $x_1$ and is therefore better represented on the low-dimensional latent space $\mathcal{S}^1$. In Fig. 1b, the stiffness matrix $\boldsymbol{X} \in \mathcal{S}^3_{++}$ of a robot controller is optimized to push objects lying on a table, with $\mathcal{S}^d_{++}$ the manifold of $d \times d$ symmetric positive definite (SPD) matrices. In this case, the stiffness along the vertical axis $x_3$ does not influence the robot's ability to push the objects. We may thus optimize the stiffness along the axes $x_1$ and $x_2$, i.e., in the latent space $\mathcal{S}^2_{++}$. Therefore, similarly to high-dimensional BO frameworks where a Euclidean latent space of the Euclidean parameter space is exploited, the objective functions may be efficiently represented in a latent space that inherits the geometry of the original Riemannian manifold. In general, this latent space is unknown and may not be aligned with the coordinate axes.

Following these observations, this paper proposes a novel high-dimensional geometry-aware BO framework (hereinafter called HD-GaBO) for optimizing parameters lying on low-dimensional Riemannian manifolds embedded in high-dimensional spaces. Our approach is based on a geometry-aware surrogate model that learns both a mapping onto a latent space inheriting the geometry of the original space, and the representation of the objective in this latent space (see § 3). The next query point is then selected on the low-dimensional Riemannian manifold using geometry-aware optimization methods. We evaluate the performance of HD-GaBO on various benchmark functions and show that it efficiently and reliably optimizes high-dimensional objective functions that feature an intrinsic low dimensionality (see § 4). Potential applications of our approach are discussed in § 5.

## 2 Background

**Riemannian Manifolds** In machine learning, diverse types of data do not belong to a vector space and thus the use of classical Euclidean methods for treating and analyzing these variables is inadequate. A common example is unit-norm data, widely used to represent directions and orientations, that can be represented as points on the surface of a hypersphere. More generally, many data are normalized in a preprocessing step to discard superfluous scaling and hence are better explained through spherical representations [19]. Notably, spherical representations have been recently exploited to design

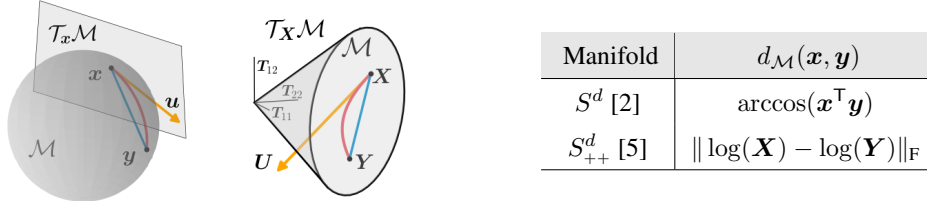

| Manifold | $d_{\mathcal{M}}(\boldsymbol{x}, \boldsymbol{y})$ |
|---|---|
| $\mathcal{S}^d$ [2] | $\arccos(\boldsymbol{x}^{\top}\boldsymbol{y})$ |
| $\mathcal{S}^d_{++}$ [5] | $\|\log(\boldsymbol{X}) - \log(\boldsymbol{Y})\|_{\mathrm{F}}$ |

Figure 2: Illustrations of the manifolds $\mathcal{S}^2$ (*left*) and $\mathcal{S}^2_{++}$ (*middle*). *Left*: Points on the surface of the sphere, such as $\boldsymbol{x}$ and $\boldsymbol{y}$ belong to the manifold. *Middle*: One point corresponds to a matrix $\left(\begin{smallmatrix} T_{11} & T_{12} \\ T_{12} & T_{22} \end{smallmatrix}\right) \in \mathrm{Sym}^2$ in which the manifold is embedded. For both graphs, the shortest path between $\boldsymbol{x}$ and $\boldsymbol{y}$ is the geodesic represented as a red curve, which differs from the Euclidean path depicted in blue. $\boldsymbol{u}$ lies on the tangent space of $\boldsymbol{x}$. The *right* table describes the distance operations on $\mathcal{S}^d$ and $\mathcal{S}^d_{++}$.

variational autoencoders [58, 12]. SPD matrices are also extensively used: They coincide with the covariance matrices of multivariate distributions and are employed as descriptors in many applications, such as computer vision [56] and brain-computer interface classification [8]. SPD matrices are also widely used in robotics in the form of stiffness and inertia matrices, controller gains, manipulability ellipsoids, among others.

Both the sphere and the space of SPD matrices can be endowed with a Riemannian metric to form Riemannian manifolds. Intuitively, a Riemannian manifold $\mathcal{M}$ is a mathematical space for which each point locally resembles a Euclidean space. For each point $\boldsymbol{x} \in \mathcal{M}$, there exists a tangent space $\mathcal{T}_{\boldsymbol{x}}\mathcal{M}$ equipped with a smoothly-varying positive definite inner product called a Riemannian metric. This metric permits us to define curve lengths on the manifold. These curves, called geodesics, are the generalization of straight lines on the Euclidean space to Riemannian manifolds, as they represent the minimum length curves between two points in $\mathcal{M}$. Fig. 2 illustrates the two manifolds considered in this paper and details the corresponding distance operations. The unit sphere $\mathcal{S}^d$ is a $d$-dimensional manifold embedded in $\mathbb{R}^{d+1}$. The tangent space $\mathcal{T}_x\mathcal{S}^d$ is the hyperplane tangent to the sphere at $\boldsymbol{x}$. The manifold of $d \times d$ SPD matrices $\mathcal{S}^d_{++}$, endowed here with the Log-Euclidean metric [5], can be represented as the interior of a convex cone embedded in its tangent space $\mathrm{Sym}^d$. Supplementary manifold operations used to optimize acquisition functions in HD-GaBO are detailed in Appendix A.

**Geometry-aware Bayesian Optimization**   The geometry-aware BO (GaBO) framework [30] aims at finding a global maximizer (or minimizer) of an unknown objective function $f$, so that $\boldsymbol{x}^* = \mathrm{argmax}_{\boldsymbol{x} \in \mathcal{X}} f(\boldsymbol{x})$, where the design space of parameters $\mathcal{X}$ is a Riemannian manifold or a sub-space of a Riemannian manifold, i.e. $\mathcal{X} \subseteq \mathcal{M}$. With GaBO, geometry-awareness is first brought into BO by modeling the unknown objective function $f$ with a GP adapted to manifold-valued data. This is achieved by defining geometry-aware kernels measuring the similarity of the parameters on $\mathcal{M}$. In particular, the geodesic generalization of the SE kernel is given by $k(\boldsymbol{x}_i, \boldsymbol{x}_j) = \theta \exp(-\beta d_{\mathcal{M}}(\boldsymbol{x}_i, \boldsymbol{x}_j)^2)$, where $d_{\mathcal{M}}(\cdot, \cdot)$ denotes the Riemannian distance between two observations and the parameters $\beta$ and $\theta$ control the horizontal and vertical scale of the function [31]. For manifolds that are not isometric to a Euclidean space, this kernel is valid, i.e. positive definite, only for parameters values $\beta > \beta_{\min}$ [18], where $\beta_{\min}$ can be determined experimentally [17, 30]. Other types of kernels are available for specific manifolds and may also be used in BO (see e.g., [45, 18, 25]).

Secondly, the selection of the next query point $\boldsymbol{x}_{n+1}$ is achieved by optimizing the acquisition function on the manifold $\mathcal{M}$. To do so, optimization algorithms on Riemannian manifolds are exploited [2]. These geometry-aware algorithms reformulate constrained problems as an unconstrained optimization on manifolds and consider the intrinsic structure of the space of interest. Also, they tend to show lower computational complexity and better numerical properties [29].

## 3   High-Dimensional Geometry-aware Bayesian Optimization

In this section, we present the high-dimensional geometry-aware BO (HD-GaBO) framework that naturally handles the case where the design space of parameters $\mathcal{X}$ is (a subspace of) a high-dimensional Riemannian manifold, i.e. $\mathcal{X} \subseteq \mathcal{M}^D$. We assume here that the objective function satisfies the low-dimensional assumption (i.e., some dimensions of the original parameter space

do not influence its value) and thus only varies within a low-dimensional latent space. Moreover, we assume that this latent space can be identified as a low-dimensional Riemannian manifold $\mathcal{M}^d$ inheriting the geometry of the original manifold $\mathcal{M}^D$, with $d \ll D$. Notice that the same assumption is generally made by Euclidean high-dimensional BO frameworks, as the objective function is represented in a latent space $\mathbb{R}^d$ of $\mathbb{R}^D$. In particular, we model the objective function $f : \mathcal{M}^D \to \mathbb{R}$ as a composition of a structure-preserving mapping $m : \mathcal{M}^D \to \mathcal{M}^d$ and a function $g : \mathcal{M}^d \to \mathbb{R}$, so that $f = g \circ m$. A model of the objective function is thus available in the latent space $\mathcal{M}^d$, which is considered as the optimization domain to maximize the acquisition function. As the objective function can be evaluated only in the original space $\mathcal{M}^D$, the query point $\boldsymbol{z} \in \mathcal{Z}$, with $\mathcal{Z} \subseteq \mathcal{M}^d$, obtained by the acquisition function is projected back into the high-dimensional manifold with the right-inverse projection mapping $m^\dagger : \mathcal{M}^d \to \mathcal{M}^D$.

In HD-GaBO, the latent spaces are obtained via nested approaches on Riemannian manifolds featuring parametric structure-preserving mappings $m : \mathcal{M}^D \to \mathcal{M}^d$. Moreover, the parameters $\boldsymbol{\Theta}_m$ and $\boldsymbol{\Theta}_g$ of the mapping $m$ and function $g$ are determined jointly in a supervised manner using a geometry-aware GP model, as detailed in § 3.1. Therefore, the observed values of the objective function are exploited not only to design the BO surrogate model, but also to drive the dimensionality reduction process towards expressive latent spaces for a data-efficient high-dimensional BO. Considering nested approaches also allows us to build a mapping $m^\dagger$ that can be viewed as the pseudo-inverse of the mapping $m$. As explained in § 3.3, the corresponding set of parameters $\boldsymbol{\Theta}_{m^\dagger}$ includes the projection mapping parameters $\boldsymbol{\Theta}_m$ and a set of reconstruction parameters $\boldsymbol{\Theta}_r$, so $\boldsymbol{\Theta}_{m^\dagger} = \{\boldsymbol{\Theta}_m, \boldsymbol{\Theta}_r\}$. Therefore, the parameters $\boldsymbol{\Theta}_r$ are determined as to minimize the reconstruction error, as detailed in § 3.2. Similarly to GaBO [30], geometry-aware kernel functions are used in HD-GaBO (see § 3.1), and the acquisition function is optimized using techniques on Riemannian manifolds, although the optimization is carried out on the latent Riemannian manifold in HD-GaBO. The proposed HD-GaBO framework is summarized in Algorithm 1.

---

**Algorithm 1:** HD-GaBO

**Input:** Initial observations $\mathcal{D}_0 = \{(\boldsymbol{x}_i, y_i)\}_{i=1}^{N_0}$, $\boldsymbol{x}_i \in \mathcal{M}^D$, $y_i \in \mathbb{R}$
**Output:** Final recommendation $\boldsymbol{x}_N$

1 **for** $n = 0, 1 \ldots, N$ **do**
2      Update the hyperparameters $\{\boldsymbol{\Theta}_m, \boldsymbol{\Theta}_g\}$ of the geometry-aware mGP model ;
3      Project the observed data into the latent space, so that $\boldsymbol{z}_i = m(\boldsymbol{x}_i)$ ;
4      Select the next query point $\boldsymbol{z}_{n+1} \in \mathcal{M}^d$ by optimizing the acquisition function in the latent space, i.e.,
        $\boldsymbol{z}_{n+1} = \mathrm{argmax}_{\boldsymbol{z} \in \mathcal{Z}}\, \gamma_n(\boldsymbol{z}; \{(\boldsymbol{z}_i, y_i)\})$ ;
5      Update the hyperparameters $\boldsymbol{\Theta}_{m^\dagger}$ of the pseudo-inverse projection ;
6      Obtain the new query point $\boldsymbol{x}_{n+1} = m^\dagger(\boldsymbol{z}_{n+1})$ in the original space ;
7      Query the objective function to obtain $y_{n+1}$ ;
8      Augment the set of observed data $\mathcal{D}_{n+1} = \{\mathcal{D}_n, (\boldsymbol{x}_{n+1}, y_{n+1})\}$ ;
9 **end**

---

## 3.1 HD-GaBO Surrogate Model

The choice of latent spaces is crucial for the efficiency of HD-GaBO as it determines the search space for the selection of the next query point $\boldsymbol{x}_{n+1}$. In this context, it is desirable to base the latent-space learning process not only on the distribution of the observed parameters $\boldsymbol{x}_n$ in the original space, but also on the quality of the corresponding values $y_n$ of the objective function. Therefore, we propose (*i*) to supervisedly learn a structure-preserving mapping onto a low-dimensional latent space, and (*ii*) to learn the representation of the objective function in this latent space along with the corresponding mapping. To do so, we exploit the so-called manifold Gaussian process (mGP) model introduced in [10]. It is important to notice that the term *manifold* denotes here a latent space, whose parameters are learned by the mGP, which does not generally correspond to a Riemannian manifold.

In a mGP, the regression process is considered as a composition $g \circ m$ of a parametric projection $m$ onto a latent space and a function $g$. Specifically, a mGP is defined as a GP so that $f \sim \mathcal{GP}(\mu_m, k_m)$ with mean function $\mu_m : \mathcal{X} \to \mathbb{R}$ and positive-definite covariance function $k_m : \mathcal{X} \times \mathcal{X} \to \mathbb{R}$ defined as $\mu_m(\boldsymbol{x}) = \mu(m(\boldsymbol{x}))$ and $k_m(\boldsymbol{x}_i, \boldsymbol{x}_j) = k(m(\boldsymbol{x}_i), m(\boldsymbol{x}_j))$, with $\mu : \mathcal{Z} \to \mathbb{R}$ and $k : \mathcal{Z} \times \mathcal{Z} \to \mathbb{R}$ a kernel function. The mGP parameters are estimated by maximizing the marginal likelihood of the model, so that $\{\boldsymbol{\Theta}_m^*, \boldsymbol{\Theta}_g^*\} = \mathrm{argmax}_{\boldsymbol{\Theta}_m, \boldsymbol{\Theta}_g}\, p(\boldsymbol{y}|\boldsymbol{X}, \boldsymbol{\Theta}_m, \boldsymbol{\Theta}_g)$.

In mGP [10], the original and latent spaces are subspaces of Euclidean spaces, so that $\mathcal{X} \subseteq \mathbb{R}^D$ and $\mathcal{Z} \subseteq \mathbb{R}^d$, respectively. Note that the idea of jointly learning a projection mapping and a representation of the objective function with a mGP was also exploited in the context of high-dimensional BO in [42]. In [10, 42], the mapping $m : \mathbb{R}^D \to \mathbb{R}^d$ was represented by a neural network. However, in the HD-GaBO framework, the design parameter space $\mathcal{X} \subseteq \mathcal{M}^D$ is a high-dimensional Riemannian manifold and we aim at learning a geometry-aware latent space $\mathcal{Z} \subseteq \mathcal{M}^d$ that inherits the geometry of $\mathcal{X}$. Thus, we define a structure-preserving mapping $m : \mathcal{M}^D \to \mathcal{M}^d$ as a nested projection from a high- to a low-dimensional Riemannian manifold of the same type, as described in § 3.3. Moreover, as in GaBO, we use a geometry-aware kernel function $k$ that allows the GP to properly measure the similarity between parameters $z = m(x)$ lying on the Riemannian manifold $\mathcal{M}^d$. Therefore, the surrogate model of HD-GaBO is a geometry-aware mGP, that leads to a geometry-aware representation of the objective function in a locally optimal low-dimensional Riemannian manifold $\mathcal{M}^d$.

Importantly, the predictive distribution for the mGP $f \sim \mathcal{GP}(\mu_m, k_m)$ at test input $\tilde{x}$ is equivalent to the predictive distribution of the GP $g \sim \mathcal{GP}(\mu, k)$ at test input $\tilde{z} = m(\tilde{x})$. Therefore, the predictive distribution can be straightforwardly computed in the latent space. This allows the optimization function to be defined and optimized in the low-dimensional Riemannian manifold $\mathcal{M}^d$ instead of the original high-dimensional parameter space $\mathcal{M}^D$. Then, the selected next query point $z_{n+1}$ in the latent space needs to be projected back onto $\mathcal{M}^D$ in order to evaluate the objective function.

## 3.2 Input Reconstruction from the Latent Embedding to the Original Space

After optimizing the acquisition function, the selected query point $z_{n+1}$ in the latent space needs to be projected back onto the manifold $\mathcal{M}^D$ in order to evaluate the objective function. For solving this problem in the Euclidean case, Moriconi et al. [42] proposed to learn a reconstruction mapping $r : \mathbb{R}^d \to \mathbb{R}^D$ based on multi-output GPs. In contrast, we propose here to further exploit the nested structure-preserving mappings in order to project the selected query point back onto the original manifold. As shown in § 3.3, a right-inverse parametric projection $m^\dagger : \mathcal{M}^d \to \mathcal{M}^D$ can be built from the nested Riemannian manifold approaches. This pseudo-inverse mapping depends on a set of parameters $\Theta_{m^\dagger} = \{\Theta_m, \Theta_r\}$. Note that the parameters $\Theta_m$ are learned with the mGP surrogate model, but we still need to determine the reconstruction parameters $\Theta_r$. While the projection mapping $m$ aimed at finding an optimal representation of the objective function, the corresponding pseudo-inverse mapping $m^\dagger$ should (ideally) project the data $z$ lying on the latent space $\mathcal{M}^d$ onto their original representation $x$ in the original space $\mathcal{M}^D$. Therefore, the parameters $\Theta_r$ are obtained by minimizing the sum of the squared residuals on the manifold $\mathcal{M}^D$, so that

$$\Theta_r^* = \underset{\Theta_r}{\arg\min} \sum_{i=1}^{n} d_{\mathcal{M}^D}^2 \big( x_i, m^\dagger(z_i; \Theta_m, \Theta_r) \big). \tag{1}$$

## 3.3 Nested Manifolds Mappings

As mentioned previously, the surrogate model of HD-GaBO learns to represent the objective function in a latent space $\mathcal{M}^d$ inheriting the geometry of the original space $\mathcal{M}^D$. To do so, the latent space is obtained via nested approaches, which map a high-dimensional Riemannian manifold to a low-dimensional latent space inheriting the geometry of the original Riemannian manifold. While various other dimensionality reduction techniques have been proposed on Riemannian manifolds [20, 52, 53, 28, 46], the resulting latent space is usually formed by curves on the high-dimensional manifold $\mathcal{M}^D$. This would still require to optimize the acquisition function on $\mathcal{M}^D$ with complex constraints, which may not be handled efficiently by optimization algorithms. In contrast, nested manifold mappings reduce the dimension of the search space in a systematic and structure-preserving manner, so that the acquisition function can be efficiently optimized on a low-dimensional Riemannian manifold with optimization techniques on Riemannian manifolds. Moreover, intrinsic latent spaces may naturally be encoded with nested manifold mappings in various applications (see Fig. 1). Nested mappings for the sphere and SPD manifolds are presented in the following.

**Sphere manifold**  The concept of nested spheres, introduced in [32], is illustrated in Fig. 3. Given an axis $v \in \mathcal{S}^D$, the sphere is first rotated so that $v$ aligns with the origin, typically defined as the north pole $(0, \ldots, 0, 1)^\top$. Then, the data $x \in \mathcal{S}^D$ (in purple) are projected onto the subsphere $\mathcal{A}^{D-1}$ defined as $\mathcal{A}^{D-1}(v, r) = \{w \in \mathcal{S}^D : d_{\mathcal{S}^D}(v, w) = r\}$, where $r \in (0, \pi/2]$, so that

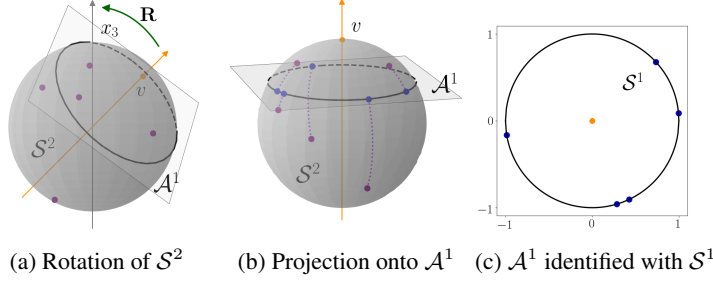

(a) Rotation of $\mathcal{S}^2$     (b) Projection onto $\mathcal{A}^1$     (c) $\mathcal{A}^1$ identified with $\mathcal{S}^1$

Figure 3: Illustration of the nested sphere projection mapping. Data on the sphere $\mathcal{S}^2$, depicted by purple dots, are projected onto the subsphere $\mathcal{A}^1$, which is then identified with the sphere $\mathcal{S}^1$.

$x_D = \cos(r)$. The last coordinate of $\boldsymbol{x}$ is then discarded and the data $\boldsymbol{z} \in \mathcal{S}^{D-1}$ (in blue) are obtained by identifying the subsphere $\mathcal{A}^{D-1}$ of radius $\sin(r)$ with the nested unit sphere $\mathcal{S}^{D-1}$ via a scaling operation. Specifically, given an axis $\boldsymbol{v}_D \in \mathcal{S}^D$ and a distance $r_D \in (0, \pi/2]$, the projection mapping $m_D : \mathcal{S}^D \to \mathcal{S}^{D-1}$ is computed as

$$\boldsymbol{z} = m_D(\boldsymbol{x}) = \underbrace{\frac{1}{\sin(r_D)}}_{\text{scaling}} \underbrace{\boldsymbol{R}_{\text{trunc}}}_{\text{rotation + dim. red.}} \underbrace{\left( \frac{\sin(r_D)\boldsymbol{x} + \sin\left(d_{\mathcal{S}^D}(\boldsymbol{v}_D, \boldsymbol{x}) - r_D\right)\boldsymbol{v}_D}{\sin\left(d_{\mathcal{S}^D}(\boldsymbol{v}_D, \boldsymbol{x})\right)} \right)}_{\text{projection onto } \mathcal{A}^{D-1}}, \quad (2)$$

with $d_{\mathcal{S}^D}$ defined as in the table of Fig. 2, $\boldsymbol{R} \in \mathrm{SO}(D)$ is the rotation matrix that moves $\boldsymbol{v}$ to the origin on the manifold and $\boldsymbol{R}_{\text{trunc}}$ the matrix composed of the $D-1$ first rows of $\boldsymbol{R}$. Notice also that the order of the projection and rotation operations is interchangeable. In (2), the data are simultaneously rotated and reduced after being projected onto $\mathcal{A}^{D-1}$. However, the same result may be obtained by projecting the rotated data $\boldsymbol{R}\boldsymbol{x}$ onto $\mathcal{A}^{D-1}$ using the rotated axis $\boldsymbol{R}\boldsymbol{v}$ and multiplying the obtained vector by the truncated identity matrix $\boldsymbol{I}_{\text{trunc}} \in \mathbb{R}^{D-1 \times D}$. This fact will be later exploited to define the SPD nested mapping. Then, the full projection mapping $m : \mathcal{S}^D \to \mathcal{S}^d$ is defined via successive mappings (2), so that $m = m_{d+1} \circ \ldots \circ m_{D-1} \circ m_D$, with parameters $\{\boldsymbol{v}_D, \ldots \boldsymbol{v}_{d+1}, r_D, \ldots r_{d+1}\}$ such that $\boldsymbol{v}_k \in \mathcal{S}^k$ and $r_k \in (0, \pi/2]$. Importantly, notice that the distance $d_{\mathcal{S}^d}(m(\boldsymbol{x}_i), m(\boldsymbol{x}_j))$ between two points $\boldsymbol{x}_i, \boldsymbol{x}_j \in \mathcal{S}^D$ projected onto $\mathcal{S}^d$ is invariant w.r.t the distance parameters $\{r_D, \ldots r_{d+1}\}$ (see Appendix B for a proof). Therefore, when using distance-based kernels, the parameters set of the mGP projection mapping corresponds to $\boldsymbol{\Theta}_m = \{\boldsymbol{v}_D, \ldots \boldsymbol{v}_{d+1}\}$. The mGP parameters optimization is thus carried out with techniques on Riemannian manifolds on the domain $\mathcal{S}^D \times \cdots \times \mathcal{S}^{d+1} \times \mathcal{M}_g$, where $\mathcal{M}_g$ is the space of GP parameters $\boldsymbol{\Theta}_g$ (usually $\mathcal{M}_g \sim \mathbb{R} \times \ldots \times \mathbb{R}$).

As shown in [32], an inverse transformation $m_D^{-1} : \mathcal{S}^{D-1} \to \mathcal{S}^D$ can be computed as

$$\boldsymbol{x} = m_D^{-1}(\boldsymbol{z}) = \boldsymbol{R}^{\mathsf{T}} \begin{pmatrix} \sin(r_{d+1})\boldsymbol{z} \\ \cos(r_{d+1}) \end{pmatrix}. \quad (3)$$

Therefore, the query point selected by the acquisition function in the latent space can be projected back onto the original space with the inverse projection mapping $m^{\dagger} : \mathcal{S}^d \to \mathcal{S}^D$ given by $m^{\dagger} = m_D^{\dagger} \circ \ldots \circ m_{d+1}^{\dagger}$. As the axes parameters are determined within the mGP model, the set of reconstruction parameters is given by $\boldsymbol{\Theta}_r = \{r_D, \ldots, r_{d+1}\}$.

**SPD manifold**    Although not explicitly named as such, the dimensionality reduction technique for the SPD manifold introduced in [26, 27] can be understood as a nested manifold mapping. Specifically, Harandi et al. [26, 27] proposed a projection mapping $m : \mathcal{S}_{++}^D \to \mathcal{S}_{++}^d$, so that

$$\boldsymbol{Z} = m(\boldsymbol{X}) = \boldsymbol{W}^{\mathsf{T}} \boldsymbol{X} \boldsymbol{W}, \quad (4)$$

with $\boldsymbol{W} \in \mathbb{R}^{D \times d}$. Note that the matrix $\boldsymbol{Z} \in \mathcal{S}_{++}^d$ is guaranteed to be positive definite if $\boldsymbol{W}$ has a full rank. As proposed in [26, 27], this can be achieved, without loss of generality, by imposing orthogonality constraint on $\boldsymbol{W}$ such that $\boldsymbol{W} \in \mathcal{G}_{D,d}$, i.e., $\boldsymbol{W}^{\mathsf{T}}\boldsymbol{W} = \boldsymbol{I}$, where $\mathcal{G}_{D,d}$ denotes the Grassmann manifold corresponding to the space of $d$-dimensional subspaces of $\mathbb{R}^D$ [15]. Therefore, in the case of the SPD manifold, the projection mapping parameter set is $\boldsymbol{\Theta}_m = \{\boldsymbol{W}\}$. Specifically, the mGP parameters are optimized on the product of Riemannian manifolds $\mathcal{G}^{D,d} \times \mathcal{M}_g$. Also, the optimization of the mGP on the SPD manifold can be simplified as shown in Appendix C.

In order to project the query point $\boldsymbol{Z} \in \mathcal{S}_{++}^d$ back onto the original space $\mathcal{S}_{++}^D$, we propose to build an inverse projection mapping based on $m$. It can be easily observed that using the pseudo-inverse $\boldsymbol{W}$ so

that $\boldsymbol{X} = \boldsymbol{W}^{\dagger^\top}\boldsymbol{Z}\boldsymbol{W}^\dagger$ does not guarantee the recovered matrix $\boldsymbol{X}$ to be positive definite. Therefore, we propose a novel inverse mapping inspired by the nested sphere projections. To do so, we observe that an analogy can be drawn between the mappings (2) and (4). Namely, the mapping (4) first consists of a rotation $\boldsymbol{R}^\top\boldsymbol{X}\boldsymbol{R}$ of the data $\boldsymbol{X} \in \mathcal{S}_{++}^D$ with $\boldsymbol{R}$ a rotation matrix whose $D$ first columns equal $\boldsymbol{W}$, i.e., $\boldsymbol{R} = (\boldsymbol{W} \quad \boldsymbol{V})$, where $\boldsymbol{W}$ can been understood as $\boldsymbol{R}_{\text{trunc}}$ in Eq. (2). Similarly to the nested sphere case, the rotated data can be projected onto a subspace of the manifold $\mathcal{S}_{++}^D$ by fixing their last coordinates. Therefore, the subspace is composed of matrices $\begin{pmatrix} \boldsymbol{W}^\top\boldsymbol{X}\boldsymbol{W} & \boldsymbol{C} \\ \boldsymbol{C}^\top & \boldsymbol{B} \end{pmatrix}$, where $\boldsymbol{B} \in \mathcal{S}_{++}^{D-d}$ is a constant matrix. Finally, this subspace may be identified with $\mathcal{S}_{++}^d$ by multiplying the projected matrix $\begin{pmatrix} \boldsymbol{W}^\top\boldsymbol{X}\boldsymbol{W} & \boldsymbol{C} \\ \boldsymbol{C}^\top & \boldsymbol{B} \end{pmatrix}$ with a truncated identity matrix $\boldsymbol{I}_{\text{trunc}} \in \mathbb{R}^{D \times d}$. Therefore, the mapping (4) is equivalently expressed as $\boldsymbol{Z} = m(\boldsymbol{X}) = \boldsymbol{I}_{\text{trunc}}^\top \begin{pmatrix} \boldsymbol{W}^\top\boldsymbol{X}\boldsymbol{W} & \boldsymbol{C} \\ \boldsymbol{C}^\top & \boldsymbol{B} \end{pmatrix} \boldsymbol{I}_{\text{trunc}} = \boldsymbol{W}^\top\boldsymbol{X}\boldsymbol{W}$.

From the properties of block matrices with positive block-diagonal elements, the projection is positive definite if and only if $\boldsymbol{W}^\top\boldsymbol{X}\boldsymbol{W} \geq \boldsymbol{C}\boldsymbol{B}\boldsymbol{C}^\top$ [6]. This corresponds to defining the side matrix as $\boldsymbol{C} = (\boldsymbol{W}^\top\boldsymbol{X}\boldsymbol{W})^{\frac{1}{2}}\boldsymbol{K}\boldsymbol{B}^{\frac{1}{2}}$, where $\boldsymbol{K} \in \mathbb{R}^{d \times D-d}$ is a contraction matrix, so that $\|\boldsymbol{K}\| \leq 1$ [6]. Based on the aforementioned equivalence, the inverse mapping $m^\dagger : \mathcal{S}_{++}^d \to \mathcal{S}_{++}^D$ is given by

$$\boldsymbol{X} = m^\dagger(\boldsymbol{Z}) = \boldsymbol{R} \begin{pmatrix} \boldsymbol{Z} & \boldsymbol{Z}^{\frac{1}{2}}\boldsymbol{K}\boldsymbol{B}^{\frac{1}{2}} \\ \boldsymbol{B}^{\frac{1}{2}}\boldsymbol{K}^\top\boldsymbol{Z}^{\frac{1}{2}} & \boldsymbol{B} \end{pmatrix} \boldsymbol{R}^\top, \tag{5}$$

with reconstruction parameters $\boldsymbol{\Theta}_r = \{\boldsymbol{V}, \boldsymbol{K}, \boldsymbol{B}\}$. The optimization (1) is thus carried out on the product of manifolds $\mathcal{G}_{D-d,d} \times \mathbb{R}^{d,D-d} \times \mathcal{S}_{++}^{D-d}$ subject to $\|\boldsymbol{K}\| \leq 1$ and $\boldsymbol{W}^\top\boldsymbol{V} = \boldsymbol{0}$. The latter condition is necessary for $\boldsymbol{R}$ to be a valid rotation matrix. We solve this optimization problem with the augmented Lagrangian method on Riemannian manifolds [38].

## 4   Experiments

In this section, we evaluate the proposed HD-GaBO framework to optimize high-dimensional functions that lie on an intrinsic low-dimensional space. We consider benchmark test functions defined on a low-dimensional manifold $\mathcal{M}^d$ embedded in a high-dimensional manifold $\mathcal{M}^D$. Therefore, the test functions are defined as $f : \mathcal{M}^D \to \mathbb{R}$, so that $y = f(m(\boldsymbol{x}))$ with $m : \mathcal{M}^D \to \mathcal{M}^d$ being the nested projection mapping, as defined in Section 3.3. The projection mapping parameters are randomly set for each trial. The search space corresponds to the complete manifold for $\mathcal{S}^D$ and to SPD matrices with eigenvalues $\lambda \in [0.001, 5]$ for $\mathcal{S}_{++}^D$. We carry out the optimization by running 30 trials with random initialization. Both GaBO and HD-GaBO use the geodesic generalization of the SE kernel and their acquisition functions are optimized using trust region on Riemannian manifolds [1] (see Appendix D). The other state-of-the-art approaches use the classical SE kernel and the constrained acquisition functions are optimized using sequential least squares programming [34]. All the tested methods use EI as acquisition function and are initialized with 5 random samples. The GP parameters are estimated using MLE. All the implementations employ GPyTorch [22], BoTorch [7] and Pymanopt [55]. Source code is available at `https://github.com/NoemieJaquier/GaBOtorch`. Supplementary results are presented in Appendix F.

In the case of the sphere manifold $\mathcal{S}^D$, we compare HD-GaBO against GaBO, the Euclidean BO and three high-dimensional BO approaches, namely dropout BO [36], SIR-BO [59], and REMBO [57], which carry out all the operations in the Euclidean space. The optimization of the acquisition function of each Euclidean BO method was adapted to fulfill the constraint $\|\boldsymbol{x}\| = 1$. Other approaches, such as the MGPC-BO of [42], are not considered here due to the difficulty of adapting them when the parameters lie on Riemannian manifolds. We minimize the Rosenbrock, Ackley, and product-of-sines functions (see also Appendix E) defined on the low-dimensional manifold $\mathcal{S}^5$ embedded in $\mathcal{S}^{50}$. Fig. 4a- 4c display the median of the logarithm of the simple regret along 300 BO iterations and the distribution of the logarithm of the BO recommendation $\boldsymbol{x}_N$ for the three functions. We observe that HD-GaBO generally converges fast and provides good optimizers for all the test cases. Moreover, it outperforms all the other BO methods for the product-of-sines function: it provides fast convergence and better optimizer with low variance. In contrast, SIR-BO, which leads to the best optimizer for the Rosenbrock function, performs poorly to optimize the product-of-sines function. Similarly, dropout achieves a similar performance as HD-GaBO for the Ackley function, but it is outperformed by HD-GaBO in the two other test cases. Moreover, it is worth noticing that GaBO converges faster to

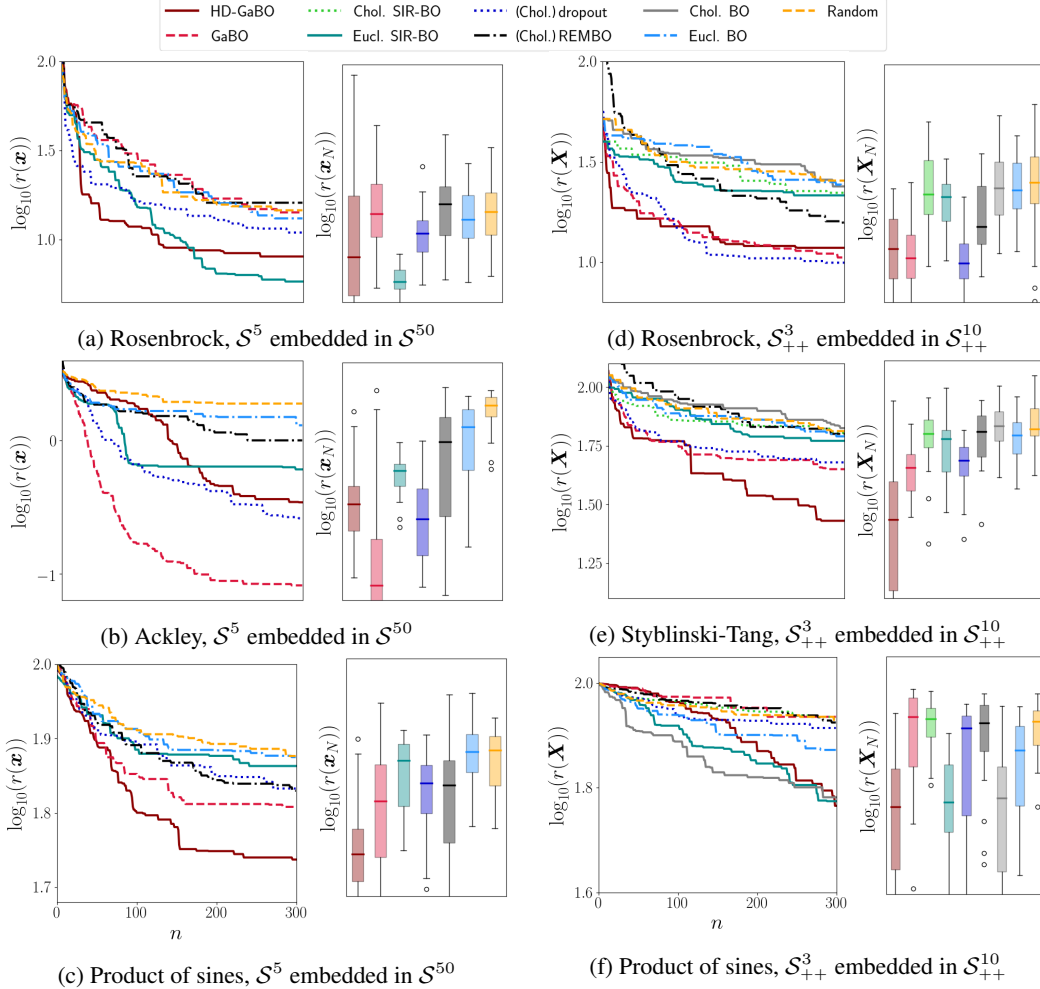

Legend:
HD-GaBO — Chol. SIR-BO — (Chol.) dropout — Chol. BO — Random — GaBO — Eucl. SIR-BO — (Chol.) REMBO — Eucl. BO

(a) Rosenbrock, $\mathcal{S}^5$ embedded in $\mathcal{S}^{50}$

(d) Rosenbrock, $\mathcal{S}^3_{++}$ embedded in $\mathcal{S}^{10}_{++}$

(b) Ackley, $\mathcal{S}^5$ embedded in $\mathcal{S}^{50}$

(e) Styblinski-Tang, $\mathcal{S}^3_{++}$ embedded in $\mathcal{S}^{10}_{++}$

(c) Product of sines, $\mathcal{S}^5$ embedded in $\mathcal{S}^{50}$

(f) Product of sines, $\mathcal{S}^3_{++}$ embedded in $\mathcal{S}^{10}_{++}$

Figure 4: Logarithm of the simple regret for benchmark test functions over 30 trials. The *left* graphs show the evolution of the median for the BO approaches and the random search baseline. The *right* graphs display the distribution of the logarithm of the simple regret of the BO recommendation $\boldsymbol{x}_N$ after 300 iterations. The boxes extend from the first to the third quartiles and the median is represented by a horizontal line. Supplementary results are provided in Appendix F.

the best optimizer than the other approaches for the Ackley function and performs better than all the geometry-unaware approaches for the product-of-sines function. This highlights the importance of using geometry-aware approaches for optimizing objective functions lying on Riemannian manifolds.

Regarding the SPD manifold $\mathcal{S}^D_{++}$, we compare HD-GaBO against GaBO, the Euclidean BO and SIR-BO (augmented with the constraint $\lambda_{\min} > 0$). Moreover, we consider alternative implementations of BO, dropout, SIR-BO and REMBO that exploit the Cholesky decomposition of an SPD matrix $\boldsymbol{A} = \boldsymbol{L}\boldsymbol{L}^\top$, so that the resulting parameter is the vectorization of the lower triangular matrix $\boldsymbol{L}$ (hereinafter denoted as Cholesky-methods). Note that we do not consider here the Euclidean version of the dropout and REMBO methods due to the difficulty of optimizing the acquisition function in the latent space while satisfying the constraint $\lambda_{\min} > 0$ for the query point in the high-dimensional manifold. We minimize the Rosenbrock, Styblinski-Tang, and product-of-sines functions defined on the low-dimensional manifold $\mathcal{S}^3_{++}$ embedded in $\mathcal{S}^{10}_{++}$. The corresponding results are displayed in Fig. 4d-4f (in logarithm scale). We observe that HD-GaBO consistently converges fast and provides good optimizers for all the test cases. Moreover, it outperforms all the other approaches for the Styblinski-Tang function. Similarly to the sphere cases, some methods are still competitive with respect to HD-GaBO for some of the test functions but perform poorly in other cases. Interestingly, GaBO performs well for both Rosenbrock and Styblinski-Tang functions. Moreover, the Euclidean

BO methods generally perform poorly compared to their Cholesky equivalences, suggesting that, although they do not account for the manifold geometry, Cholesky-based approaches provide a better representation of the SPD parameter space than the Euclidean methods.

## 5 Potential Applications

After evaluating the performance of HD-GaBO in various benchmark artificial landscapes, we discuss potential real-world applications of the proposed approach. First, HD-GaBO may be exploited for the optimization of controller parameters in robotics. Of particular interest is the optimization of the error gain matrix $\boldsymbol{Q}_t \in \mathcal{S}_{++}^{D_x}$ and control gain matrix $\boldsymbol{R}_t \in \mathcal{S}_{++}^{D_u}$ in linear quadratic regulators (LQR), where $D_x$ and $D_u$ are the dimensionality of the system state and control input, respectively. The system state may consist of the linear and angular position and velocity of the robot end-effector, so that $D_x = 13$, and $D_u$ corresponds to Cartesian accelerations or wrench commands. Along some parts of the robot trajectory, the error w.r.t. some dimensions of the state space may not influence the execution of the task, i.e., affect negligibly the LQR cost function. Therefore, the matrix $\boldsymbol{Q}_t$ for this trajectory segment may be efficiently optimized in a latent space $\mathcal{S}_{++}^{d_x}$ with $d_x < D_x$. A similar analysis applies for $\boldsymbol{R}$. Notice that, although BO has been applied to optimize LQR parameters [40, 41], the problem was greatly simplified as only diagonal matrices $\boldsymbol{Q}$ and $\boldsymbol{R}$ were considered in the optimization, resulting in a loss of flexibility in the controller. From a broader point of view, the low-dimensional assumption may also apply in the optimization of gain matrices for other types of controllers.

Another interesting application is the identification of dynamic model parameters of (highly-) redundant robots. These parameters typically include the inertia matrix $\boldsymbol{M} \in \mathcal{S}_{++}^{D}$ with $D$ being the number of robot joints. As discussed in [60], a low-dimensional representation of the parameter space and state-action space may be sufficient to determine the system dynamics. Therefore, the inertia matrix may be more efficiently represented and identified in a lower-dimensional SPD latent space.

In the context of directional statistics [54, 47], HD-GaBO may be used to adapt mixtures of von Mises-Fisher distributions, whose mean directions belong to $\mathcal{S}^D$. On a different topic, object shape spaces are typically characterized on high-dimensional unit spheres $\mathcal{S}^D$. Several works have shown that the main features of the shapes are efficiently represented in a low-dimensional latent space $\mathcal{S}^d$ inheriting the geometry of the original manifold (see e.g., [32]. Therefore, such latent spaces may be exploited for shape representation optimization. Along a similar line, skeletal models, which seek at capturing the interior of objects, lie on a Cartesian product of manifolds that involves the unit hypersphere [48]. The relevant data structure is efficiently expressed in a product of low-dimensional manifolds of the same types, so that HD-GaBO may be exploited to optimize skeletal models.

## 6 Conclusion

In this paper, we proposed HD-GaBO, a high-dimensional geometry-aware Bayesian optimization framework that exploited geometric prior knowledge on the parameter space to optimize high-dimensional functions lying on low-dimensional latent spaces. To do so, we used a geometry-aware GP that jointly learned a nested structure-preserving mapping and a representation of the objective function in the latent space.We also considered the geometry of the latent space while optimizing the acquisition function and took advantage of the nested mappings to express the next query point in the high-dimensional parameter space. We showed that HD-GaBO not only outperformed other BO approaches in several settings, but also consistently performed well while optimizing various objective functions, unlike geometry-unaware state-of-the-art methods.

An open question, shared across various high-dimensional BO approaches, concerns the model dimensionality mismatch. In order to avoid suboptimal solutions where the optimum of the function may not be included in the estimated latent space, we hypothesize that the dimension $d$ should be selected slightly higher in case of uncertainty on its value [35]. A limitation of HD-GaBO is that it depends on nested mappings that are specific to each Riemannian manifold. Therefore, such mappings may not be available for all kinds of manifolds. Also, the inverse map does not necessarily exist if the manifold contains self-intersection. In this case, a non-parametric reconstruction mapping may be learned (e.g., based on wrapped GP [39]). However, most of the Riemannian manifolds encountered in machine learning and robotics applications do not self-intersect, so that this problem is avoided. Future work will investigate the aforementioned aspects.

## Broader Impact

The HD-GaBO formulation presented in this paper makes a step towards more explainable and interpretable BO approaches. Indeed, in addition to the benefits in terms of performance, the inclusion of domain knowledge via Riemannian manifolds into the BO framework permits to treat the space parameters in a principled way. This can notably be contrasted with approaches based on random features, that generally remain hard to interpret for humans. As often, the gains in terms of explainability and interpretability come at the expense of the low computational cost that characterizes random-based approaches. However, the carbon footprint of the proposed approach remains low compared to many deep approaches used nowadays in machine learning applications.

## Acknowledgments and Disclosure of Funding

This work was mainly developed during a PhD sabbatical at the Bosch Center for Artificial Intelligence (Renningen, Germany). This work was also partially supported by the FNS/DFG project TACT-HAND, as part of the PhD thesis of the first author, carried out at the Idiap Research Institute (Martigny, Switzerland), while also affiliated to the Ecole Polytechnique Fédérale de Lausanne (Lausanne, Switzerland). Noémie Jaquier is now affiliated with the Karlsruhe Institute of Technology (Karlsruhe, Germany).

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
