[Supplementary Material]

# High-Dimensional Bayesian Optimization via Nested Riemannian Manifolds: Supplementary Material

**Noémie Jaquier**[1,2]
[1]Idiap Research Institute
1920 Martigny, Switzerland
noemie.jaquier@kit.edu

**Leonel Rozo**[2]
[2]Bosch Center for Artificial Intelligence
71272 Renningen, Germany
leonel.rozo@de.bosch.com

## Abstract

This document provides (*A*) a supplementary background on Riemannian manifolds, (*B*) a proof that the distance between two points on a nested hypersphere manifold is independent w.r.t. the parameters $\{r_D, \ldots r_{d+1}\}$, (*C*) an approximation of the SPD distance for the mGP kernel, (*D*) the algorithm for trust region on Riemannian manifolds used for optimizing acquisition functions in HD-GaBO, (*E*) the equations of the benchmark functions used in the experiments of the main paper, and (*F*) supplementary results.

# Appendices

## A    Supplementary Background on Riemannian Manifolds

Optimization algorithms on Riemannian manifolds used in this paper to optimize the acquisition function in a geometry-aware manner, have been developed by taking advantage of the Euclidean tangent space $\mathcal{T}_{\boldsymbol{x}}\mathcal{M}$ linked to each point $\boldsymbol{x}$ on the manifold $\mathcal{M}$. To utilize the Euclidean tangent spaces, we need mappings back and forth between $\mathcal{T}_{\boldsymbol{x}}\mathcal{M}$ and $\mathcal{M}$, which are known as exponential and logarithmic maps. The exponential map $\mathrm{Exp}_{\boldsymbol{x}} : \mathcal{T}_{\boldsymbol{x}}\mathcal{M} \to \mathcal{M}$ maps a point $\boldsymbol{u}$ in the tangent space of $\boldsymbol{x}$ to a point $\boldsymbol{y}$ on the manifold, so that it lies on the geodesic starting at $\boldsymbol{x}$ in the direction $\boldsymbol{u}$ and such that the geodesic distance $d_{\mathcal{M}}$ between $\boldsymbol{x}$ and $\boldsymbol{y}$ is equal to norm of $\boldsymbol{u}$. The inverse operation is called the logarithmic map $\mathrm{Log}_{\boldsymbol{x}} : \mathcal{M} \to \mathcal{T}_{\boldsymbol{x}}\mathcal{M}$. Notice that these different operations are determined based on the Riemannian metric with which the manifold is endowed.

The exponential and logarithmic maps related to hypersphere manifolds can be found, e.g., in [2]. In the case of the SPD manifold, several Riemannian metrics have been proposed in the literature, notably the affine-invariant [10] and Log-Euclidean [3] metrics, which both set matrices with null or negative eigenvalues at an infinite distance of any SPD matrix. The exponential and logarithmic maps based on the two aforementioned metrics can be found in the corresponding publications. Detailed explanations on several SPD metrics can also be found in [11]. While the affine-invariant metric provides excellent theoretical properties, it is computationally expensive in practice, therefore leading to a need for simpler metrics. In this context, the Log-Euclidean metric has been shown to perform well in a variety of applications.

# B   Distances between Points on Nested Spheres

The geometry-aware mGP used in HD-GaBO involves the computation of kernel functions based on distances between data projected onto nested Riemannian manifolds with the projection mapping $m : \mathcal{S}^D \to \mathcal{S}^d$. We compute here the distance between projected data on nested spheres and show that this distance is invariant to the parameters $\{r_D, \ldots r_{d+1}\}$.

To do so, we first compute the distance $d_{\mathcal{S}^{D-1}}(m_D(\boldsymbol{x}_i), m_D(\boldsymbol{x}_j))$ between two points $\boldsymbol{x}_i, \boldsymbol{x}_j \in \mathcal{S}^D$ projected onto $\mathcal{S}^{D-1}$. Given an axis $\boldsymbol{v}_D \in \mathcal{S}^D$ and a distance $r_D \in \,]0, \pi/2]$, the projection mapping $m_D : \mathcal{S}^D \to \mathcal{S}^{D-1}$ is computed as Eq.2 of the main paper

$$\boldsymbol{z} = m_D(\boldsymbol{x}) = \underbrace{\frac{1}{\sin(r_D)}}_{\text{scaling}} \; \underbrace{\boldsymbol{I}_{\text{trunc}} \boldsymbol{R}}_{\text{dim. red. + rot.}} \; \underbrace{\left( \frac{\sin(r_D)\boldsymbol{x} + \sin\left(d_{\mathcal{S}^D}(\boldsymbol{v}_D, \boldsymbol{x}) - r_D\right)\boldsymbol{v}_D}{\sin\left(d_{\mathcal{S}^D}(\boldsymbol{v}_D, \boldsymbol{x})\right)} \right)}_{\text{projection onto } \mathcal{A}^{D-1}}, \tag{1}$$

where $\boldsymbol{I}_{\text{trunc}}$ is the $D-1 \times D$ truncated identity matrix. By exploiting the identity

$$\sin(\alpha - \beta) = \sin(\alpha)\cos(\beta) - \cos(\alpha)\sin(\beta), \tag{2}$$

and the distance formula $d_{\mathcal{S}^D}(\boldsymbol{v}_D, \boldsymbol{x}) = \arccos(\boldsymbol{v}_D^\mathsf{T}\boldsymbol{x})$, we can further rewrite (1) as

$$\boldsymbol{z} = m_D(\boldsymbol{x}) = \underbrace{\frac{1}{\sin(r_D)}}_{\text{scaling}} \; \underbrace{\boldsymbol{I}_{\text{trunc}} \boldsymbol{R}}_{\text{dim. red. + rot.}} \; \underbrace{\left( \frac{\sin(r_D)}{\sin\left(d_{\mathcal{S}^D}(\boldsymbol{v}_D, \boldsymbol{x})\right)}(\boldsymbol{x} + \boldsymbol{v}_D^\mathsf{T}\boldsymbol{x}\boldsymbol{v}_D) + \cos(r_D)\boldsymbol{v}_D \right)}_{\text{projection onto } \mathcal{A}^{D-1}}. \tag{3}$$

The distance $d_{\mathcal{S}^{D-1}}\left(m_D(\boldsymbol{x}_i), m_D(\boldsymbol{x}_j)\right)$ is given by

$$d_{\mathcal{S}^{D-1}}\left(m_D(\boldsymbol{x}_i), m_D(\boldsymbol{x}_j)\right) = d_{\mathcal{S}^{D-1}}(\boldsymbol{z}_i, \boldsymbol{z}_j) = \arccos(\boldsymbol{z}_i^\mathsf{T}\boldsymbol{z}_j). \tag{4}$$

By defining the projection onto $\mathcal{A}^{D-1}$ as the function $\boldsymbol{z} = p(\boldsymbol{x})$, we can compute

$$\boldsymbol{z}_i^\mathsf{T}\boldsymbol{z}_j = \frac{1}{\sin^2(r_D)} p(\boldsymbol{x}_i)^\mathsf{T} \boldsymbol{R}^\mathsf{T} \boldsymbol{I}_{\text{trunc}}^\mathsf{T} \boldsymbol{I}_{\text{trunc}} \boldsymbol{R} \, p(\boldsymbol{x}_j), \tag{5}$$

$$= \frac{1}{\sin^2(r_D)} \left( p(\boldsymbol{x}_i)^\mathsf{T} \boldsymbol{R}^\mathsf{T} \boldsymbol{R} \, p(\boldsymbol{x}_j) - \cos^2(r_D) \right), \tag{6}$$

$$= \frac{1}{\sin^2(r_D)} \left( p(\boldsymbol{x}_i)^\mathsf{T} p(\boldsymbol{x}_j) - \cos^2(r_D) \right), \tag{7}$$

$$= \frac{1}{\sin^2(r_D)} \left( \frac{\sin^2(r_D)(\boldsymbol{x}_i - \boldsymbol{v}_D^\mathsf{T}\boldsymbol{x}_i\boldsymbol{v}_D)^\mathsf{T}(\boldsymbol{x}_j - \boldsymbol{v}_D^\mathsf{T}\boldsymbol{x}_j\boldsymbol{v}_D)}{\sin\left(d_{\mathcal{S}^D}(\boldsymbol{v}_D, \boldsymbol{x}_i)\right)\sin\left(d_{\mathcal{S}^D}(\boldsymbol{v}_D, \boldsymbol{x}_j)\right)} + \cos^2(r_D)\boldsymbol{v}_D^\mathsf{T}\boldsymbol{v}_D - \cos^2(r_D) \right), \tag{8}$$

$$= \frac{(\boldsymbol{x}_i - \boldsymbol{v}_D^\mathsf{T}\boldsymbol{x}_i\boldsymbol{v}_D)^\mathsf{T}(\boldsymbol{x}_j - \boldsymbol{v}_D^\mathsf{T}\boldsymbol{x}_j\boldsymbol{v}_D)}{\sin\left(d_{\mathcal{S}^D}(\boldsymbol{v}_D, \boldsymbol{x}_i)\right)\sin\left(d_{\mathcal{S}^D}(\boldsymbol{v}_D, \boldsymbol{x}_j)\right)}, \tag{9}$$

so that $\boldsymbol{z}_i^\mathsf{T}\boldsymbol{z}_j$, and thus the distance (4), are invariant w.r.t. $r_D$. Note that (6) was obtained by using the fact that the last coordinate of the projections $\boldsymbol{R} \, p(\boldsymbol{x}_i)$ and $\boldsymbol{R} \, p(\boldsymbol{x}_j)$ is equal to $\cos(r_D)$ from the nested sphere mapping definition. We then used the rotation matrix property $\boldsymbol{R}^\mathsf{T}\boldsymbol{R} = \boldsymbol{I}$ to obtain (7) and the unit-norm property of $\boldsymbol{v}_D$, so that $\boldsymbol{v}_D^\mathsf{T}\boldsymbol{v}_D = 1$ to obtain (9).

As the distance (4) is invariant w.r.t. $r_D$ for any dimension $D$ and as the mapping $m$ is a composition of successive mappings $m_D$, we can straightforwardly conclude that the distance $d_{\mathcal{S}^d}\left(m(\boldsymbol{x}_i), m(\boldsymbol{x}_j)\right)$ with $\boldsymbol{x}_i, \boldsymbol{x}_j \in \mathcal{S}^D$ and $d \le D$ is invariant w.r.t. the parameters $\{r_D, \ldots r_{d+1}\}$.

## C  Approximation of the SPD distance for the mGP kernel

In [7], the SE kernel based on the affine-invariant SPD distance

$$d_{\mathcal{S}_{++}^d}(\boldsymbol{X}, \boldsymbol{Y}) = \|\log(\boldsymbol{X}^{-\frac{1}{2}} \boldsymbol{Y} \boldsymbol{X}^{-\frac{1}{2}})\|_{\mathrm{F}},$$

was used for GaBO on the SPD manifold. During the GP parameters optimization in GaBO, the distances between each pair of SPD data only depend on the data and are solely computed at the beginning of the optimization process. In contrast, in HD-GaBO, the distances between the projected SPD data vary as a function of $\boldsymbol{W}$ and therefore must be computed at each optimization step. This results in a computationally expensive optimization of the mGP parameters. In order to alleviate this computational burden, we propose to use the SE kernel based on the Log-Euclidean SPD distance [3]

$$d_{\mathcal{S}_{++}^d}(\boldsymbol{X}_i, \boldsymbol{X}_j) = \|\log(\boldsymbol{X}_i) - \log(\boldsymbol{X}_j)\|_{\mathrm{F}}.$$

Moreover, as shown in [6], we can approximate $\log(\boldsymbol{W}^\mathsf{T} \boldsymbol{X} \boldsymbol{W}) \simeq \boldsymbol{W}^\mathsf{T} \log(\boldsymbol{X}) \boldsymbol{W}$, so that

$$d_{\mathcal{S}_{++}^d}(\boldsymbol{W}^\mathsf{T} \boldsymbol{X}_i \boldsymbol{W}, \boldsymbol{W}^\mathsf{T} \boldsymbol{X}_j \boldsymbol{W}) \simeq \|\boldsymbol{W}^\mathsf{T} \left(\log(\boldsymbol{X}_i) - \log(\boldsymbol{X}_j)\right) \boldsymbol{W}\|_{\mathrm{F}}. \tag{10}$$

Therefore, the difference between the logarithm of SPD matrices is fixed throughout the optimization process. This allows us to optimize the mGP parameters at a lower computational cost without affecting consequently the performance of HD-GaBO. Note that the Log-Euclidean based SE kernel is positive definite for all the values of the parameter $\beta$ [8].

## D  Optimization of Acquisition Functions: Trust Region on Riemannian Manifolds

In this paper, we exploit trust-region (TR) methods on Riemannian manifolds, as introduced in [1], to optimizing the acquisition function $\gamma_n$ in the latent space at each iteration $n$ of HD-GaBO. The recursive process of the TR methods on Riemannian manifolds, described in Algorithm 1, involves the same steps as its Euclidean equivalence, namely: (*i*) the optimization of a quadratic subproblem $m_k$ trusted locally, i.e., in a region around the iterate (step 3); (*ii*) the update of the trust-region parameters — typically the trust-region radius $\Delta_k$ — (steps 5-11); (*iii*) the iterate update, where a candidate is accepted or rejected in function of the quality of the model $m_k$ (steps 12-16). The differences with the Euclidean version are:

1. The trust-region subproblem given by

$$\underset{\boldsymbol{\eta} \in \mathcal{T}_{\boldsymbol{z}_k} \mathcal{M}}{\arg\min}\, m_k(\boldsymbol{\eta}) \text{ s.t. } \|\boldsymbol{\eta}\|_{\boldsymbol{z}_k} \leq \Delta_k, \tag{11}$$

$$\text{with } m_k(\boldsymbol{\eta}) = \phi_n(\boldsymbol{z}_k) + \langle -\nabla \phi_n(\boldsymbol{z}_k), \boldsymbol{\eta} \rangle_{\boldsymbol{z}_k} + \frac{1}{2} \langle \boldsymbol{H}_k, \boldsymbol{\eta} \rangle_{\boldsymbol{z}_k}, \tag{12}$$

   is defined and solved in the tangent space $\mathcal{T}_{\boldsymbol{z}_k} \mathcal{M}$, with $\nabla \phi_n(\boldsymbol{z}_k) \in \mathcal{T}_{\boldsymbol{z}_k} \mathcal{M}$ and $\boldsymbol{H}_k$ some symmetric operator on $\mathcal{T}_{\boldsymbol{z}_k} \mathcal{M}$. Therefore, its solution $\boldsymbol{\eta}_k$ corresponds to the projection of the next candidate in the tangent space of the iterate $\boldsymbol{z}_k$. A truncated CG algorithm to solve the subproblem is provided in Algorithm 2.

2. As a consequence of the previous point, the candidate is obtained by computing $\mathrm{Exp}_{\boldsymbol{z}_k}(\boldsymbol{\eta}_k)$.

The symmetric operator $\boldsymbol{H}_k$ on the tangent space $\mathcal{T}_{\boldsymbol{z}_k} \mathcal{M}$ typically approximates the Riemannian Hessian $\mathrm{Hess}\,\phi_n(\boldsymbol{z}_k)[\boldsymbol{\eta}]$, which may be expensive to compute. For example, one may use the approximation of the Hessian with finite difference approximation introduced in [4], that has been shown to retain global convergence of the Riemannian TR algorithm. Also notice that the steps 4 and 11 of Algorithm 2 correspond to solving the second-order equation

$$\langle \boldsymbol{\nu}_j, \boldsymbol{\nu}_j \rangle_{\boldsymbol{z}_k} + 2\tau_\Delta \langle \boldsymbol{\nu}_j, \boldsymbol{\delta}_j \rangle_{\boldsymbol{z}_k} + \tau_\Delta^2 \langle \boldsymbol{\delta}_j, \boldsymbol{\delta}_j \rangle_{\boldsymbol{z}_k} = \Delta_k^2, \tag{13}$$

for $\tau_\Delta$, which was obtained from $\|\boldsymbol{\nu}_j + \tau_\Delta \boldsymbol{\delta}_j\|_{\boldsymbol{z}_k} = \Delta_k$ by using the relationship between the norm and the inner product and the properties of inner products.

**Algorithm 1:** Optimization of acquisition function with trust region on Riemannian manifolds

**Input:** Acquisition function $\gamma_n$, initial iterate $\boldsymbol{z}_0 \in \mathcal{M}$, maximal trust radius $\Delta_{\max} > 0$, initial trust radius $\Delta_0 < \Delta_{\max}$, acceptance threshold $\rho$

**Output:** Next parameter point $\boldsymbol{x}_{n+1}$

1  Set $\phi_n = -\gamma_n$ as the function to minimize ;
2  **for** $k = 0, 1 \dots, K$ **do**
3     Compute the candidate $\mathrm{Exp}_{\boldsymbol{z}_k}(\boldsymbol{\eta}_k)$ by solving the subproblem

$$\boldsymbol{\eta}_k = \underset{\boldsymbol{\eta} \in \mathcal{T}_{\boldsymbol{z}_k}\mathcal{M}}{\mathrm{argmin}}\; m_k(\boldsymbol{\eta}) \text{ s.t. } \|\boldsymbol{\eta}\|_{\boldsymbol{z}_k} \leq \Delta_k,$$

     with $m_k(\boldsymbol{\eta}) = \phi_n(\boldsymbol{z}_k) + \langle -\nabla\phi_n(\boldsymbol{z}_k), \boldsymbol{\eta}\rangle_{\boldsymbol{z}_k} + \frac{1}{2}\langle \boldsymbol{H}_k, \boldsymbol{\eta}\rangle_{\boldsymbol{z}_k}$ (Algo. 2);

4     Evaluate the accuracy of the model by computing $\rho_k = \frac{\phi_n(\boldsymbol{z}_k) - \phi_n\left(\mathrm{Exp}_{\boldsymbol{z}_k}(\boldsymbol{\eta}_k)\right)}{m_k(\boldsymbol{0}) - m_k(\boldsymbol{\eta}_k)}$;

5     **if** $\rho_k < \frac{1}{4}$ **then**
6       Reduce the trust radius $\Delta_{k+1} = \frac{1}{4}\Delta_k$ ;
7     **else if** $\rho_k > \frac{3}{4}$ *and* $\|\boldsymbol{\eta}_k\|_{\boldsymbol{z}_k} = \Delta_k$ **then**
8       Expand the trust radius $\Delta_{k+1} = \min(2\Delta_k, \Delta_{\max})$;
9     **else**
10      $\Delta_{k+1} = \Delta_k$ ;
11     **end**
12     **if** $\rho_k > \rho$ **then**
13       Accept the candidate and set $\boldsymbol{z}_{k+1} = \mathrm{Exp}_{\boldsymbol{z}_k}(\boldsymbol{\eta}_k)$ ;
14     **else**
15       Reject the candidate and set $\boldsymbol{z}_{k+1} = \boldsymbol{z}_k$ ;
16     **end**
17     **if** *a convergence criterion is reached* **then**
18       break
19     **end**
20  **end**
21  Set $\boldsymbol{x}_{n+1} = \boldsymbol{z}_{k+1}$

For the cases where the domain of HD-GaBO needs to be restricted to a subspace of the manifold, we propose to extend the TR algorithm to cope with linear constraints. Similarly to the Euclidean case [5, 12], the trust-region subproblem can be augmented as

$$\underset{\boldsymbol{\eta} \in \mathcal{T}_{\boldsymbol{z}_k}\mathcal{M}}{\mathrm{argmin}}\; m_k(\boldsymbol{\eta}) \text{ s.t. } \|\boldsymbol{\eta}\|_{\boldsymbol{z}_k} \leq \Delta_k^2 \text{ and } \|(\boldsymbol{c}_k + \nabla\boldsymbol{c}_k^{\mathsf{T}}\boldsymbol{\eta})^-\|_{\boldsymbol{z}_k} \leq \xi_k, \tag{14}$$

where $\boldsymbol{c}_k$ is a vector of linearized constraints $\boldsymbol{c}_k = (c_1(\boldsymbol{z}_k) \dots c_M(\boldsymbol{z}_k))^{\mathsf{T}}$, $\nabla\boldsymbol{c}_k$ is the corresponding gradient, $(x)^- = x$ for equality constraints $c_m(\boldsymbol{z}_k) = 0$ and $(x)^- = \min(0, x)$ for inequality constraints $c_m(\boldsymbol{z}_k) \geq 0$. The subproblem (14) can be solved with the augmented Lagrangian or the exact penalty methods on Riemannian manifolds presented in [9].

In the context of Bayesian optimization, a common assumption is that the optimum should not lie in the border of the search space. Therefore, the acquisition function does not need to be exactly maximized close to the border of the search space. However, it is important to stay in the search space to cope with physical limits or safety constraints of the system. By exploiting these two considerations, we propose to optimize the subproblem (14) in a simplified way, by adapting Algorithm 2 to cope with the constraints. At each iteration, we verify that the iterate $\boldsymbol{\nu}_{j+1} = \boldsymbol{\nu}_j + \alpha_j\boldsymbol{\delta}_j$ satisfies the constraints. If the constraints are not satisfied, the value of the step size $\alpha_j$ is adjusted and the algorithm is terminated. This process is described in Algorithm 3 and is used to augment the steps 5, 12 and 14 of Algorithm 2. Note that the proposed approach ensures that the constraints are satisfied, but is not guaranteed to converge to optima lying on a constraint border. However, we did not observe any significant difference in the performance of HD-GaBO by using this approach compared to more sophisticated methods.

**Algorithm 2:** Truncated conjugate gradient for solving the trust-region subproblem (step 3 of Algorithm 1)

**Input:** Trust-region subproblem 11 to minimize, given $\phi_n(z_k)$, $H_k$
**Output:** Update vector $\eta_k$

1   Set the initial iterate $\nu_0 = 0$, residual $r_0 = \nabla\phi_n(z_k)$ and search direction $\delta_0 = -r_0$;
2   **for** $j = 0, 1\ldots, J$ **do**
3     **if** $\langle\delta_j, H_k\delta_j\rangle_{z_k} \leq 0$ **then**
4       Compute $\tau_\Delta \geq 0$ s.t. $\|\nu_j + \tau_\Delta\delta_j\|_{z_k} = \Delta_k$ ;
5       Set $\nu_{j+1} = \nu_j + \tau_\Delta\delta_j$ ;
6       break
7     **end**
8     Compute the step size $\alpha_j = \frac{\langle r_j, r_j\rangle_{z_k}}{\langle\delta_j, H_k\delta_j\rangle_{z_k}}$ ;
9     Set $\nu_{j+1} = \nu_j + \alpha_j\delta_j$ ;
10    **if** $\|\nu_{j+1}\|_{z_k} \geq \Delta_k$ **then**
11      Compute $\tau_\Delta \geq 0$ s.t. $\|\nu_j + \tau_\Delta\delta_j\|_{z_k} = \Delta_k$ ;
12      Set $\nu_{j+1} = \nu_j + \tau_\Delta\delta_j$ ;
13      break
14    **end**
15    Set $r_{j+1} = r_j + \alpha_j H_k\delta_j$;
16    Set $\delta_{j+1} = -r_{j+1} + \frac{\langle r_{j+1}, r_{j+1}\rangle_{z_k}}{\langle r_j, r_j\rangle_{z_k}}\delta_j$ ;
17    **if** *a convergence criterion is reached* **then**
18      break
19    **end**
20 **end**
21 Set $\eta_k = \nu_{j+1}$

---

**Algorithm 3:** Addition to steps 5, 12 and 14 of Algorithm 2 to solve the trust-region subproblem (14).

Set $c_k = c(z_k)$ ;
**if** $\|(c_k + \nabla c_k^\top\nu_{j+1})^-\|_{z_k} \geq 0$ **then**
   Compute $\tau_c \geq 0$ s.t. $\|\left(c_k + \nabla c_k^\top(\nu_j + \tau_c\delta_j)\right)^-\|_{z_k} = 0$;
   Set $\nu_{j+1} = \nu_j + \tau_c\delta_j$ ;
   break
**end**

## E   Benchmark Test Functions

This appendix gives the equations of the benchmark test functions considered in the experiment section of the main paper. Namely, we minimize the Ackley, Rosenbrock, Styblinski-Tang and product-of-sines functions defined as

$$f_{\text{Ackley}}(x) = -20\exp\left(-0.2\sqrt{\frac{1}{d}\sum_{i=1}^{d}x_i^2}\right) - \exp\left(\frac{1}{d}\sum_{i=1}^{d}\cos(2\pi x_i)\right) + 20 + \exp(1),$$

$$f_{\text{Rosenbrock}}(x) = \sum_{i=1}^{d-1}\left(100(x_{i+1} - x_i^2)^2 + (x_i - 1)^2\right),$$

$$f_{\text{Styblinski-Tang}}(x) = \frac{1}{2}\sum_{i=1}^{d}\left((5x_i)^4 - 16(5x_i)^2 + 5(5x_i)\right),$$

$$f_{\text{product-of-sines}}(x) = 100\sin(x_1)\prod_{i=1}^{d}\sin(x_i).$$

Figure 1: Logarithm of the simple regret for benchmark test functions over 30 trials. The *left* graphs show the evolution of the median for the BO approaches and the random search baseline. The *right* graphs display the distribution of the logarithm of the simple regret of the BO recommendation $\boldsymbol{x}_N$ after 300 iterations. The boxes extend from the first to the third quartiles and the median is represented by a horizontal line.

# F   Supplementary Results

The aim of this appendix is to complement the results presented in the main paper. The experiments presented in this section were carried out in the same conditions as in the main paper. For the sphere manifold $\mathcal{S}^D$, we minimize the Rosenbrock, Ackley, and product-of-sines functions defined on the low-dimensional manifold $\mathcal{S}^5$ embedded in $\mathcal{S}^{70}$. Fig. 1a- 1c display the median of the logarithm of the simple regret along 300 BO iterations and the distribution of the logarithm of the BO recommendation $\boldsymbol{x}_N$ for the three functions. Regarding the SPD manifold $\mathcal{S}_{++}^D$, we minimize the Rosenbrock, Styblinski-Tang, and product-of-sines functions defined on the low-dimensional manifold $\mathcal{S}_{++}^3$ embedded in $\mathcal{S}_{++}^{12}$. The corresponding results are displayed in Fig. 1d-1f (in logarithm scale). The results presented in this appendix support the analysis drawn in the experiment section of the main paper and validate the use of HD-GaBO for original manifolds of higher dimensionality. Namely, we observe that HD-GaBO consistently converges fast and provides good optimizers for all the test cases. Moreover, it outperforms all the other approaches for the product-of-sines function on the sphere manifold and for the Styblinski-Tang function on the SPD manifold. Also, some methods are still competitive with respect to HD-GaBO for some of the test functions but perform poorly in other cases.