[Reviews · NeurIPS 2020]

Review 1

Summary and Contributions: This manuscript considers Bayesian optimization of (nominally) high-dimensional functions. This is a perennial challenge for Bayesian optimization, as many off-the-shelf models are not able to effectively cope with the curse of dimensionality, and the sample efficiency improvements of Bayesian optimization seen in lower dimensions do not necessarily carry over. The folk intuition -- that had better be true of we are to have any hope -- is that most real objective functions are not truly high dimensional, but rather vary primarily along some much lower dimensional subspace. If we could only identify that subspace, we could reduce the problem to a much easier, lower-dimensional version. This vision has inspired many previous investigations, which for example sought to exploit low-dimensional linear projections and/or additive decompositions with low-dimensional components. Here the authors seek to learn a low-dimensional Riemannian manifold embedded in the domain and transfer the optimization problem onto this intrinsically lower-dimensional space. The bulk of the paper (and the supplement) is devoted to building theoretical machinery to support this vision, with a brief empirical investigation as well showing the proposed methods have promise. After the discussion phase I am somewhat less enthusiastic about this paper as the other reviewers have convinced me that the scope of the proposed methods may be somewhat limited in practical settings. I am still positive overall but I would encourage the authors to better motivate the framework (and the decisions made along the way) in terms of foreseen applications.

Strengths: This work is a technical tour de force that manages to introduce and distill numerous deep ideas into a plausible, working system. The vision is sound and the approach is sophisticated. The approach would unquestionably be of interest to the NeurIPS community (or at least the subset interested in Bayesian optimization and Gaussian process modeling at large), and provides tools opening doors to a variety of lines of future research.

Weaknesses: Quite honestly, there is little to complain about here. Perhaps the biggest limitation of the work is a slight disconnect between the framework and actual practical applications, but this is mitigated by the fact that it is mostly a theoretical contribution introducing new modeling approaches rather than targeting a specific application. I did wish when reading through that a paragraph or two might have been devoted to some practical commentary, but with so much to cover (and the authors _very obviously_ struggling with the space limitations), I understand that it simply would not fit. However, the authors may want to consider whether a venue allowing more space might be a more effective means of communicating their message. NeurIPS offers a big audience but precious little bandwidth to convey complicated ideas. Just a thought.

Correctness: Yes, as far as I can tell.

Clarity: Ignoring the expected terseness and a couple of typos, yes. Just a couple of very minor suggestions I marked while reading through: - 52: robot ability -> robot's ability - 59: you say "high-diemensional" Riemannian manifolds here but I suppose you mean low-dimensional manifolds embedded in a high-dimensional space? - 75: They -> they - 82: permits -> permits us - 140: even if "supervisedly" is a word (and it doesn't read like one to me), it's an awkward one

Relation to Prior Work: Yes, the discussion and consideration of previous work is thorough. Just a couple more related works you can consider citing if you see fit: - Garnett, et al. Active learning of linear embeddings for Gasussian processes, UAI 2014. [also considers active sampling to learn a linear embedding from a high- to low-dimensional space] - Gardner, et al. Discovering and exploiting additive structure for Bayesian optimization, AISTATS 2017. [yet another work attempting to discover and exploit additive structure for Bayesian optimization]

Reproducibility: Yes

Additional Feedback: Just one technical comment: the inverse map m⁻¹ does not necessarily exist if the manifold contains self-intersection: consider for example an immersion of a Klein bottle in R³. Is this possibility somehow explicitly ruled out?


Review 2

Summary and Contributions: The paper describes the inclusion of Riemannian manifold geometric information in Gaussian process regression for Bayesian optimization. Two different manifolds are treated: the sphere and symmetric positive definite matrices. The approach incorporates dimension reduction. Experimental results are presented on several synthetic functions.

Strengths: The main interest is to brings novel ideas to Bayesian optimization.

Weaknesses: Only toy examples are provided, without a realistic test case such as the ones discussed in introduction (e.g., robot arms). While the results are good overall, the difference with other methods (without prior geometric information) is not so striking on several instances. The geometric structure must be known beforehand.

Correctness: Yes

Clarity: The paper is clear and pedagogical.

Relation to Prior Work: See Duvenaud, D. K. Automatic Model Construction with Gaussian Processes University of Cambridge, University of Cambridge, 2014 (for topological manifolds) Ginsbourger, D., Roustant, O., & Durrande, N. (2016). On degeneracy and invariances of random fields paths with applications in Gaussian process modelling. Journal of statistical planning and inference, 170, 117-128. Gaudrie, D., Le Riche, R., Picheny, V., Enaux, B., & Herbert, V. (2020). Modeling and optimization with Gaussian processes in reduced eigenbases. Structural and Multidisciplinary Optimization, 1-19.

Reproducibility: No

Additional Feedback: What is the effect of model misspecification, e.g., if d is not known? Adding an algorithmic summary would be beneficial. ### Post rebuttal comments ### The authors have responded to my comments. This paper has generated interesting discussions about the interest and limitations of using the appropriate geometrical information (which could be also included in the additional page of content). I thus increased my score to accept.


Review 3

Summary and Contributions: The authors consider the problem of Bayesian Optimisation in high dimensional spaces and propose a method to reduce this to a problem in lower dimension.

Strengths: The proposed technique appears to be built on sound ideas which are well motivated (from a theoretical point of view, not from an application point of view, see 'weaknesses below).

Weaknesses: The main weakness I'd say is that it is unclear what the applications of the proposed techniques are. The introduction mentions applications such as robotic arms or learning of directions, but these are all low dimensional. The experiment section does not help here since it concentrate on mostly artificial data.

Correctness: The claims appear to be correct.

Clarity: Overall yes, but please see the Additional Feedback section below for specific comments and suggestions as some terms used in the paper can be confusing.

Relation to Prior Work: Yes.

Reproducibility: Yes

Additional Feedback: * Figure 1: these examples are interesting applications in general, but are not good examples for this paper since they are not high dimensional examples. * Figure 1 again: Please define what are S^3_{++} and S^2_{++} here * What does it mean for an objective function to only vary within a low dimensional latent space? Please define this precisely in the paper. * What is a structure preserving mapping mapping between M^D and M^d? Such a map cannot preserve the Riemannian structure since its differential cannot have full rank with d < D. Please make this clearer. * Would it be possible to use another term for the map m^{-1}, since this map is not really the inverse of m (do you assume it is a right inverse? section 3.2 does suggest so)? ======================== Update after authors feedback ======================== I thank the authors for their feedback. I will maintain my score of 7 as I found the ideas in this paper interesting and overall well introduced. I maintain some reservations on the applicability of these ideas, even if high dimensional Bayesian optimisation is a hard problem in general. I would also like to point out that, in my opinion, the paragraph the authors said they would add in the introduction in too technical for an introduction.


Review 4

Summary and Contributions: This paper proposes a Bayesian optimization method called high-dimensional geometry aware Bayesian optimization (HD-GaBO) for a setting where the parameter space is high-dimensional and non-Euclidean. It uses a surrogate model that learns a mapping from a high dimensional Riemannian manifold, where the parameters lie, to a lower dimensional Riemannian manifold and a representation of the objective in this lower dimensional manifold. Update after the rebuttal - Keeping my original score since the authors have not satisfactorily answered the key concern in my original review: "We have an objective function to optimize where the optimizing parameters lie on a high dimensional Riemannian manifold and the authors use a latent space, which is again a Riemannian manifold. Problems where the optimizing parameters lie on a Riemannian manifold can naturally occur, but why does the latent space have to be a Riemannian manifold as well? Why would a Euclidean latent space be not sufficient?". From the reviewer discussions, it seemed that some other reviewers also share this concern.

Strengths: 1) Generally speaking, the idea is somewhat interesting, but it needs to be motivated and empirically backed.

Weaknesses: 1) The problem setting is very specific and hypothetical. We have an objective function to optimize where the optimizing parameters lie on a high dimensional Riemannian manifold and it somehow makes sense to use a latent space, which is again a Riemannian manifold. Problem where the optimizing parameters lie on a Riemannian manifold can naturally occur. But why does the latent space have to be a Riemannian manifold as well? Why would a Euclidean latent space be not sufficient. Figure 1 shows two somewhat abstract examples of lower dimensional manifolds as subspaces of higher dimensional manifolds of the same geometry. But could the authors elaborate more on where these kind of settings occur in real world machine learning/optimization problems? Lines 111-113 say: "We assume here that the objective function satisfies the low-dimensional assumption and thus only varies within a low-dimensional latent space. Moreover, we assume that this latent space can be identified as a low-dimensional Riemannian manifold Md inheriting the geometry of the original manifold MD, with d ≪ D" - authors should describe real-world problem settings where these assumptions make sense. 2) The experimental evaluation is not sufficient. No real-world datasets have been used. Probably because the problem setting is too specific/hypothetical to find real worl examples? (see the point (1) above). 3) The key contribution of this paper seem to be a combination of the ideas presented in [26] (Bayesian optimization on Riemannian manifolds), [35] (Bayesian optimization using low dimensional latent spaces). Therefore, the novelty of this work is not very high.

Correctness: 1) As detailed about the empirical evaluation is not sufficient to judge the merits of this work. Appropriate real-world datasets should be used.

Clarity: It is relatively well written and key points are sufficiently described.

Relation to Prior Work: Authors appropriately discuss relevant prior work such as [26] and [35].

Reproducibility: No

Additional Feedback: This problem setting is way too specific and might not be broad enough to publish in a major machine learning conference such as NeurIPS.

[Author Response · NeurIPS 2020]

We would like to thank the reviewers for their useful feedback. We first address the comment shared by all the reviewers
concerning the practical applications of our work. Then, we address other comments individually. Notice that the extra
page of content allowed in the final version will allow us to improve the paper according to your recommendations.

**Practical applications of HD-GaBO (answer to all the reviewers)**   First of all, we would like to emphasize that
our manuscript is mainly a theoretical contribution and so it aims at introducing the theoretical tools for the proposed
HD-GaBO framework. However, we agree with the reviewers that presenting and discussing potential applications
may make our motivation clearer and led to future applied extensions of our approach. Therefore we propose to
add the following paragraph in the introduction of the paper. Our latent Riemannian manifolds may be exploited
for the optimization of controller parameters in robotics. Of particular interest is the optimization of the error gain
matrix $\boldsymbol{Q}_t \in \mathcal{S}_{++}^{D_x}$ and control gain matrix $\boldsymbol{R}_t \in \mathcal{S}_{++}^{D_u}$ in linear quadratic regulators (LQR), where $D_x$ and $D_u$ are
the dimensionality of the system state and control input, respectively. The system state may consist of the linear and
angular position and velocity of the robot end-effector, so $D_x = 13$, and $D_u$ corresponds to Cartesian accelerations or
wrench commands. Along some parts of the robot trajectory, the error w.r.t. some dimensions of the state space may not
influence the execution of the task (i.e., insignificant effects on the LQR cost function), so that the matrix $\boldsymbol{Q}_t$ for this
trajectory segment may be efficiently optimized in a latent space $\mathcal{S}_{++}^{d_x}$ with $d_x < D_x$. A similar analysis applies for $\boldsymbol{R}$.
Notice that, although BO has been applied to optimize LQR parameters *[A. Marco et al. Automatic LQR tuning based on*
*Gaussian process global optimization. ICRA, 2016.]*, the problem was greatly simplified as only diagonal matrices $\boldsymbol{Q}$ and $\boldsymbol{R}$
were considered in the optimization, resulting in a loss of flexibility in the controller. From a broader point of view, the
low-dimensional assumption may also apply in the optimization of gain matrices for other types of controllers. Another
interesting application is the identification of dynamic model parameters of (highly-)redundant robots. These parameters
typically include the inertia matrix $\boldsymbol{M} \in \mathcal{S}_{++}^{D}$ with $D$ being the number of robot joints. As discussed in *[S. Zhu et al.*
*Efficient model identification for tensegrity locomotion. IROS, 2018.]*, a low-dimensional representation of the parameter space
and state-action space may be sufficient to determine the system dynamics. Therefore, the inertia matrix may be more
efficiently represented in a lower-dimensional SPD latent space. A third application concerns the optimization of object
shape representations. Indeed, shape spaces are typically characterized on high-dimensional unit spheres $\mathcal{S}^D$. Several
works have shown that the main features of the shapes are efficiently represented in a low-dimensional latent space $\mathcal{S}^d$
inheriting the geometry of the original manifold (see e.g.,*[S. Jung et al. Analysis of principal nested spheres. Biometrika,*
*99(3):551–568, 2012.]*). Therefore, such latent spaces may be exploited for shape representation optimization.

**Answers to Reviewer 1**   We appreciate the positive feedback of the reviewer. We will correct the indicated typos and
discuss the suggested related works. We agree that the inverse map does not necessarily exist if the manifold contains
self-intersection. In this case, a possibility would be to learn a non-parametric reconstruction mapping minimizing the
sum of the squared residuals, e.g., based on a wrapped Gaussian process *[A. Mallasto et al. Wrapped Gaussian Process*
*Regression on Riemannian Manifolds. CVPR, 2018.]*. However, most of the Riemannian manifolds encountered in machine
learning and robotics applications do not self-intersect, so that this problem is avoided.

**Answers to Reviewer 2**   We agree that prior knowledge on the geometric structure is necessary. However, an advantage
of HD-GaBO is its ability to exploit this prior knowledge to build a latent manifold from nested reconstructions that take
advantage of the known geometry. We will add an algorithmic summary at the beginning of Section 3 and also discuss
the suggested related works, specially that of D. Duvenaud on topological manifolds. Model dimensionality mismatch
is a common problem in many dimensionality reduction methods. Intuitively, if $d$ is unknown and estimated lower
than the real $d$, the optimum of the function may not be included in the estimated latent space, leading to a suboptimal
solution. In order to attenuate this effect, we hypothesize that the dimension $d$ should be selected slightly higher in case
of uncertainty on its value. We will investigate this point more thoroughly in our future work.

**Answers to Reviewer 3**   The examples presented in Fig. 1 aim at illustrating our assumption, as high-dimensional
parameter spaces are hard to visualize. However, these examples are extensible to higher dimensions. Concerning the
comment about what we mean with an objective function that only varies within a low-dimensional latent space, this
implies that some dimensions of the parameter space do not influence the value of the function. By structure-preserving
mapping, we understand a projection from a high-dimensional manifold to a low-dimensional manifold which has the
same geometric properties in the lower-dimensional space. We will precise these points in the paper. Moreover, we
indeed assume that the map $m^{-1}$ is a right inverse. We will rename it for sake of clarity.

**Answers to Reviewer 4**   We believe that the contribution of our paper goes beyond the combination of the ideas
presented in [26] and [35]. Indeed, the use of nested mappings is an important theoretical component of our framework
and provides an interpretable dimensionality reduction strategy in contrast to random embeddings. Moreover, although
the idea of using Euclidean latent spaces in BO is quite popular, to the best of our knowledge, latent spaces that
preserve the geometric properties of a non-Euclidean spaces have not been investigated in BO. As stated previously,
our manuscript is mainly a theoretical contribution. However, we evaluated our framework on classical benchmark
functions widely used in BO, which we projected on Riemannian manifolds. We consider the application of HD-GaBO
on complex real experiments (see applications above) as part of a future journal work (due to space constraints).

[Meta-Review · NeurIPS 2020]

All reviewers agreed on the technical quality of this paper and that the ideas are interesting. Opinions diverged on (a) the possible applications and (b) the degree to which these matter. There was extensive discussion amongst all reviewers (and I thank them for this). By the end, I think many reviewers shared a concern that the motivation for this work is a remaining weakness. I strongly recommend this be taken seriously in a final revision if the impact of this paper is to be maximised. Despite this, NeurIPS is a suitable venue for theoretical work and given this article is otherwise well written and interesting, I find it reasonable that the community can decide to what extent they are excited about this after publication.